# Identifiability and Generalizability from Multiple Experts in Inverse Reinforcement Learning

**Paul Rolland**
LIONS, EPFL
Lausanne, Switzerland
`paul.rolland@epfl.ch`

**Luca Viano**
LIONS, EPFL
Lausanne, Switzerland
`luca.viano@epfl.ch`

**Norman Schürhoff**
SFI, UNIL
Lausanne, Switzerland
`norman.schuerhoff@unil.ch`

**Boris Nikolov**
SFI, UNIL
Lausanne, Switzerland
`boris.nikolov@unil.ch`

**Volkan Cevher**
LIONS, EPFL
Lausanne, Switzerland
`volkan.cevher@epfl.ch`

## Abstract

While Reinforcement Learning (RL) aims to train an agent from a reward function in a given environment, Inverse Reinforcement Learning (IRL) seeks to recover the reward function from observing an expert's behavior. It is well known that, in general, various reward functions can lead to the same optimal policy, and hence, IRL is ill-defined. However, [1] showed that, if we observe two or more experts with different discount factors or acting in different environments, the reward function can under certain conditions be identified up to a constant. This work starts by showing an equivalent identifiability statement from multiple experts in tabular MDPs based on a rank condition, which is easily verifiable and is shown to be also necessary. We then extend our result to various different scenarios, i.e., we characterize reward identifiability in the case where the reward function can be represented as a linear combination of given features, making it more interpretable, or when we have access to approximate transition matrices. Even when the reward is not identifiable, we provide conditions characterizing when data on multiple experts in a given environment allows to generalize and train an optimal agent in a new environment. Our theoretical results on reward identifiability and generalizability are validated in various numerical experiments.

## 1 Introduction

Engineering a reward function in Reinforcement Learning can be troublesome in certain scenarios like driving [2], robotics [3], and economics/finance [4]. In economics and finance, the reward or objective/utility function of the agent are of fundamental importance but are not known a priori [5–8]. In such cases, it may be easier to get demonstrations from an expert policy. Therefore, multiple algorithms have been developed to learn from demonstrations, e.g., in inverse reinforcement learning (IRL) and imitation learning (IL).

In IRL, the goal is to recover the reward function maximized by the agent, while in IL the expert demonstrations are used solely to learn a nearly optimal policy. In economics/finance, inference on the reward function is the focus of a large literature on estimation, testing, and policy analysis of structural models [9–11]. However, the reward function is often highly parameterized and represented by a low-dimensional set of parameters, or the literature focuses on estimating reduced-form causal relationships but not the true reward function [12, 13]. The attractiveness of IRL relies on the fact that the reward function is the most "succinct" representation of a task [14]. Indeed, identifying the

36th Conference on Neural Information Processing Systems (NeurIPS 2022).

reward function for each state-action pair allows generalizing the task to different transition dynamics and environments, which is not possible when using IL or highly parameterized structural models.

However, the IRL problem is unfortunately ill-posed since there always exist infinitely many reward functions for which the observed expert policy is optimal [15, 16]. The problem is known as reward shaping, and it is intuitively explained with the fact that, in the long term, the optimal policy is not affected by inflating the reward in the current period and decreasing the one in the next. This difficulty originated a long debate on advantages and disadvantages of IL and IRL [17–20].

When multiple experts are available, differing in the transition matrices of the environments they each act in, and/or their discount factors, IRL can in certain cases infer the true reward function, up to a constant [21–23, 1]. Inspired by [1], we derive an equivalent necessary and sufficient condition on the expert environments, which is easily verifiable, ensuring that the true reward can be identified up to a constant shift. When this identifiability condition holds, the state-action dependent rewards can be recovered from expert demonstrations. We then derive identifiability results in various alternative scenarios, e.g., when we only have access to approximate transition matrices and, alternatively, when the reward function is known to be a linear combination of given features [24, 25].

However, full reward identifiability remains a strong requirement, and we provide a negative result of non-identifiability from any number of experts, in the presence of exogenous variables in the MDP. Nonetheless, even when the identifiability condition does not hold, the recovered reward function could still be used to train an optimal expert for a different environment. To this end, we characterize situations where observing multiple experts in given environments allows to train an optimal agent in a new environment.

## 2 Related work

Since its introduction in [15, 16], the IRL problem has been known to be ill-posed, since the observed expert policy can be optimal with respect to various reward functions. The set of reward transformations that preserve policy optimality are studied in [26, 16, 1, 27, 28]. [29] studied the unidentifiability related to suboptimal experts.

In this paper, we assume access to the optimal entropy regularized policies of multiple experts. Significant progress has been made to construct heuristics that select a single reward function from the set of IRL solutions (often called the feasible set), such as feature-based matching [30], maximum margin IRL [31], maximum causal entropy IRL [32, 33], maximum relative entropy IRL [34], Bayesian IRL [35–37], first-order optimality conditions [38, 39] or second-order optimality conditions [40, 41]. Popular IL algorithms implicitly select a feasible reward function via a convex reward regularizer [19, 42, 43] or using preference/ranking based algorithms [44, 45]. However, none of these approaches guarantee the identification of the true reward function.

The problem of identifiability in IRL has been investigated first in [21, 22] that study a setting where the learner can actively select optimal experts in multiple environments. The main result in [21, 22] is that interactively querying environments outputs a reward within a constant shift from the true one. The multiple experts setting has also been studied in [46] but in the context of value alignment verification where the aim is not to recover the reward function but rather verify that the value function of the agent is close to a target value. IRL from multiple MDPs also appears in [23] where the authors consider the problem of learning a reward function compatible with a dataset of demonstrations collected by multiple experts. In addition, [47] study structural conditions on the MDP for reward identification in the finite horizon setting and [48] study identifiability in linearly solvable MDPs.

Our work is inspired by [1]. Our first identifiability result provides an equivalent statement as their *value distinguishability* condition, but can be easily checked in practice, and allows to derive other identifiability results in alternative scenarios. Finally, the motivation for IRL is often predicting the expert behavior under new transitions dynamics [49, 50, 20]. We show that for this goal, it is not necessary to identify the exact reward, hence we give a condition on the observed experts' environments and the test environment under which an optimal expert can be trained in the test environment. This perspective has also been taken in [51]. However, this work requires stronger assumptions on the transfer environment that we avoid in this paper, only requiring access to multiple experts. Moreover, our work contributes to AI safety [52–54] alleviating the *reward hacking* and *side effects* problems [53]. Indeed, by restricting the reward to linear combinations of a set of chosen

features, we can provably recover an interpretable reward function inducing the optimal behavior, which is particularly desirable in medical applications [55, 56].

An important consideration for IRL comes from [57] that formalizes the fact that there exist tasks that can not be induced by optimizing a reward function. In this work and in IRL in general, we bypass this difficulty assuming that the expert is optimizing a reward function.

## 2.1 Related works in the economics literature

The economics/finance literature differentiates between axiomatic and revealed preference theory. In axiomatic preference theory, the reward function is posited or derived from basic axioms. In empirical and experimental work, however, simple reward function specifications are often rejected and agents have been shown to exhibit behavioral biases and/or non-standard preferences.

Differently, our work relates to the literature on revealed preference. Revealed preference theory, initiated by [58, 59], provides an approach to analyze actions (e.g., consumer's demand or investors' trading) by assuming that observed choices provide information about the underlying preferences, or reward function. Revealed preference theory is, hence, similar in spirit to IRL. But IRL has not widely been used in revealed preference analysis. We refer to [60, 61] for excellent reviews of recent advances in revealed preference theory. The goal of revealed preference theory is to recover the agents' preferences. This task is important because knowledge of the reward function is required to conduct counterfactual policy analysis. Notice that for this task, knowing only the policy function is insufficient. In financial applications, for instance, the impact of a Tobin tax can be assessed only knowing investors' preferences for trading (see, e.g. [62]).

## 3 Preliminaries

A typical RL environment is characterised by a Markov Decision Process $\mathcal{M} = \{\mathcal{S}, \mathcal{A}, T, \gamma, r, P_0\}$, where $\mathcal{S}, \mathcal{A}$ are the sets of states and actions respectively, $T : \mathcal{S} \times \mathcal{A} \times \mathcal{S} \to [0, 1]$ is the state transition probability, i.e., $T(s'|s, a)$ denotes the probability of arriving in state $s'$ when taking action $a$ in state $s$. $R : \mathcal{S} \times \mathcal{A} \to \mathbb{R}$ denotes the reward function, $\gamma$ the discount factor and $P_0$ is the initial state distribution. At each time step $t$, an agent observes the current state $s_t \in \mathcal{S}$ and takes an action $a_t \sim \pi(\cdot|s_t)$ where $\pi$ is the agent's policy which determines a distribution over all actions in $\mathcal{A}$ at every state. The agent gets a reward $r_t = r(s_t, a_t)$ and transitions to a new state $s_{t+1}$ sampled according to the transition probability $T$.

An agent acting optimally in $\mathcal{M}$ seeks to maximize its cumulative sum of rewards. In addition, we assume that the agent seeks to diversify its possible actions, and hence that it maximizes the following entropy regularized sum of discounted rewards:

$$V_\lambda^\pi(s) = \mathbb{E}_s^\pi \left[ \sum_{t=0}^\infty (\gamma^t (r(s_t, a_t) + \lambda \mathcal{H}(\pi(\cdot|s_t)))) \right], \tag{1}$$

where $\mathbb{E}_s^\pi$ denotes the expectation over trajectories $\{(s_t, a_t\}_{t \geq 0}$ starting from state $s_0 = s$ and following policy $\pi$ and $\mathcal{H}(\pi) = -\sum_{a \in \mathcal{A}} \pi(a) \log \pi(a)$ is the entropy of $\pi$. The function $V_\lambda^\pi$ is called the (entropy regularized) value function of $\pi$.

In Inverse RL, the reward function $r$ is unknown, but we observe an agent acting optimally with respect to some reward function, and we wish to recover the reward function that the agent optimizes. We now recall some results from [1].

**Theorem 1.** *For a fixed policy $\pi(a|s) > 0$, discount factor $\gamma \in [0, 1)$, and an arbitrary choice of function $v : \mathcal{S} \to \mathbb{R}$, there is a unique corresponding reward function*

$$r(s, a) = \lambda \log \pi(a|s) - \gamma \sum_{s' \in \mathcal{S}} T(s'|s, a)v(s') + v(s)$$

*such that the MDP with reward $r$ yields an entropy-regularized optimal policy $\pi_\lambda^* = \pi$ and $V_\lambda^\pi = v$.*

By observing a single expert, it is hence possible to design a reward that yields any arbitrary value function, and there are hence $|\mathcal{S}|$ degrees of freedom remaining in the recovered reward function. An idea explored in [1] is to assume that we observe two experts in two different MDPs with different

transition dynamics and discount rates, but acting optimally with respect to the same reward function. The authors show that the reward can be identified up to a constant from observing the expert policies provided that the MDPs of the experts satisfy the following *value-distinguishing* assumption.

**Definition 2.** *Consider a pair of Markov decision problems on the same state and action spaces, but with respective discount rates $\gamma_1, \gamma_2$ and transition probabilities $T^1, T^2$. We say that this pair is value-distinguishing if, for any function $v^1, v^2 : \mathcal{S} \to \mathbb{R}$, the statement*

$$v^1(s) - \gamma_1 \sum_{s' \in \mathcal{S}} T^1(s'|s,a)v^1(s') = v^2(s) - \gamma_2 \sum_{s' \in \mathcal{S}} T^2(s'|s,a)v^2(s') \ for \ all \ a \in \mathcal{A}, s \in \mathcal{S} \quad (2)$$

*implies at least one of $v^1$ and $v^2$ is a constant function.*

The way this assumption is stated makes it difficult to verify in practice, and the authors of [1] do not attempt to verify it in their experiments.

## 4 Reward identification and generalization

In this section, we present our main theoretical results on reward identifiability and generalizability. In the first part, we show an equivalent condition to Definition 2 for reward identification from two experts (Theorem 3). The simplicity of our condition makes it easily verifiable and extendable to various scenarios, in particular to the cases where we observe more than two experts (Corollary 5), when the class of rewards is linearly parameterized with a set of given features (Theorem 7), or when we have access to approximated transition matrices (Theorem 8). We also provide a negative result on reward non-identifiability in MDPs with exogenous variables, which are common in many real world scenarios. In the second part, we analyse reward generalizability. Here, we provide a condition guaranteeing that a reward compatible with two experts leads to an optimal policy in a third environment (Theorem 11). The proofs of the results are all postponed to Appendix A.

### 4.1 Reward identifiability

Consider two Markov decision problems on the same set of states and actions $\mathcal{S}$ and $\mathcal{A}$ respectively, but with different transition dynamics $T^1, T^2$ and discount factors $\gamma_1, \gamma_2$. Let $r \in \mathbb{R}^{|\mathcal{S}| \times |\mathcal{A}|}$ be the reward function common to the two environments, and let $v^1, v^2 \in \mathbb{R}^{|\mathcal{S}|}$ be the entropy regularized values functions associated expert policies $\pi^1$ and $\pi^2$ in each environment respectively. According to Theorem 1, we have that $\forall (s,a) \in \mathcal{S} \times \mathcal{A}$,

$$r(s,a) = \lambda \log \pi^1(a|s) - \gamma_1 \sum_{s' \in \mathcal{S}} T^1(s'|s,a)v^1(s') + v^1(s)$$

$$= \lambda \log \pi^2(a|s) - \gamma_2 \sum_{s' \in \mathcal{S}} T^2(s'|s,a)v^2(s') + v^2(s).$$

We hence deduce that $\forall a \in \mathcal{A}$,

$$\begin{pmatrix} I - \gamma_1 T_a^1 & -(I - \gamma_2 T_a^2) \end{pmatrix} \begin{pmatrix} v^1 \\ v^2 \end{pmatrix} = \lambda \log \pi^2(\cdot|a) - \lambda \log \pi^1(\cdot|a), \quad (3)$$

where $\forall a \in \mathcal{A}$, $T_a^i \in \mathbb{R}^{\mathcal{S} \times \mathcal{S}}$ is the transition matrix for action $a$ and expert $i = 1, 2$, i.e., $T_a^i(s,s') = T^i(s'|s,a)$. By including all available actions to the experts, we can write

$$\begin{pmatrix} I - \gamma_1 T_{a_1}^1 & -(I - \gamma_2 T_{a_1}^2) \\ \vdots & \vdots \\ I - \gamma_1 T_{a_{|\mathcal{A}|}}^1 & -(I - \gamma_2 T_{a_{|\mathcal{A}|}}^2) \end{pmatrix} \begin{pmatrix} v^1 \\ v^2 \end{pmatrix} = \begin{pmatrix} \lambda \log \pi^2(\cdot|a_1) - \lambda \log \pi^1(\cdot|a_1) \\ \vdots \\ \lambda \log \pi^2(\cdot|a_{|\mathcal{A}|}) - \lambda \log \pi^1(\cdot|a_{|\mathcal{A}|}) \end{pmatrix}. \quad (4)$$

In order to identify a unique reward function, we need to identify a unique associated value function. We hence want the linear system (4) to yield a unique solution, i.e., the $|\mathcal{A}||\mathcal{S}| \times 2|\mathcal{S}|$ matrix on the left hand side to be full rank, i.e., to have rank $2|\mathcal{S}|$. However, it is well known that, for any MDP, adding a constant to the reward would not change the associated optimal policy. Hence, there is an intrinsic

degree of freedom in reward identifiability which is impossible to get rid of from only observing expert policies. In order to identify the reward up to a constant, we need this degree of freedom to be the only one in the linear system (4), i.e., the associated matrix to have rank $2|\mathcal{S}| - 1$. This result is summarized in the following theorem, and its complete proof can be found in Appendix A.1.

**Theorem 3.** *Consider two Markov decision problems on the same set of states and actions, but with different transition dynamics $T_1, T_2$ and discount factors $\gamma_1, \gamma_2$. Suppose that we observe two experts acting each in one of these environments, optimally with respect to the same reward function, in the sense that their policies maximize the entropy regularized reward in their respective environments. Then, the reward function can be recovered up to the addition of a constant if and only if*

$$rank \begin{pmatrix} I - \gamma_1 T_{a_1}^1 & I - \gamma_2 T_{a_1}^2 \\ \vdots & \vdots \\ I - \gamma_1 T_{a_{|\mathcal{A}|}}^1 & I - \gamma_2 T_{a_{|\mathcal{A}|}}^2 \end{pmatrix} = 2|\mathcal{S}| - 1. \tag{5}$$

This condition turns out to be equivalent to Definition 2, as shown at the end of Appendix A.1, but is stated in a way that is easier to check in practice and allows us to further characterize identifiability in various scenarios. First of all, this result naturally extends to the case where we observe any number of experts. We provide hereafter the result in the case of three experts.

**Corollary 4.** *Consider three Markov decision problems on the same set of states and actions, but with different transition dynamics $T_1, T_2, T_3$ and discount factors $\gamma_1, \gamma_2, \gamma_3$. Suppose that we observe three experts acting each in one of these environments, optimally with respect to the same reward function. Then, the reward function can be recovered up to the addition of a constant if and only if*

$$rank \begin{pmatrix} I - \gamma_1 T_{a_1}^1 & I - \gamma_2 T_{a_1}^2 & \mathbf{0} \\ \vdots & \vdots & \vdots \\ I - \gamma_1 T_{a_{|\mathcal{A}|}}^1 & I - \gamma_2 T_{a_{|\mathcal{A}|}}^2 & \mathbf{0} \\ I - \gamma_1 T_{a_1}^1 & \mathbf{0} & I - \gamma_3 T_{a_1}^3 \\ \vdots & \vdots & \vdots \\ I - \gamma_1 T_{a_{|\mathcal{A}|}}^1 & \mathbf{0} & I - \gamma_3 T_{a_{|\mathcal{A}|}}^3 \end{pmatrix} = 3|\mathcal{S}| - 1. \tag{6}$$

An interesting scenario is the one where the two experts act in the same environment, and only the discount rate is varied.

**Corollary 5.** *Consider two Markov decision problems on the same set of states and actions, with the same transition matrix $T$ and reward function but different discount factors $\gamma_1 \neq \gamma_2$. Then, the reward function is identifiable up to a constant by observing two experts in $(T, \gamma_1), (T, \gamma_2)$ iff*

$$rank \begin{pmatrix} T_{a_1} - T_{a_2} \\ \vdots \\ T_{a_1} - T_{a_{|\mathcal{A}|}} \end{pmatrix} = |\mathcal{S}| - 1. \tag{7}$$

**Remark 1.** *Interestingly, condition (7) is equivalent to the condition for identification of a action-independent reward from a single expert, assuming such a reward exists ([1], Corollary 3).*

Next, we provide a negative result concerning MDPs with exogenous variables, i.e., a variable whose dynamics are independent of the agent's action. This MDP class is common in economics/finance and has been studied in many real world scenarios including inventory control problems [63], variable weather conditions and customer demands [64], wildfire management [65], and stock market fluctuations [66]. We also provide examples involving such variables in the experimental section.

**Corollary 6.** *Suppose that the state space is constructed as a set of variables each taking a finite number of values, i.e., $\mathcal{S} = \{s \in \mathbb{R}^d : s_i \in \mathcal{S}_i\}$. The transition matrices for each action $a$ can be defined by specifying the evolution of each state variable $s_i^{t+1}$ depending on $(s^t, a)$. Suppose that there exists a state variable whose evolution only depends on its previous value, but neither on the other state variables nor the action taken: such a variable is called an **exogenous** variable. Note that this variable can still affect the evolution of all other variables, and its evolution can vary across the environment of the observed experts. Then, the reward function is **not** identifiable (even up to a constant) using any number of experts.*

Such a negative result motivates the search for milder requirements than arbitrary reward identification, which is too hard of a goal to achieve in certain scenarios.

A possible way to improve reward identifiability is to restrict the class of possible rewards, e.g., by constraining it to be a linear combination of a set of chosen features. This is known as Feature matching IRL [49, 67–70]. The smaller the set of features, the easier to identify the reward, as described in the following theorem. This method also allows to recover a more interpretable reward function, since the recovered parameters are associated with specific features.

**Theorem 7.** *Suppose that we restrict the class of possible reward functions to the one parameterized as $r_w(s, a) = w^T f_{s,a} \ \forall a \in \mathcal{A}, s \in \mathcal{S}$ where $f : \mathcal{S} \times \mathcal{A} \to \mathbb{R}^d$ is a given feature function, and $w \in \mathbb{R}^d$ denotes the reward parameters. Suppose that the $d$ chosen features are linearly independent, i.e., that $f_{s,a}^T v = 0 \ \forall s, a \Rightarrow v = 0$. Then, if $\mathbf{1} \in Im \begin{pmatrix} f_{a_1} \\ \vdots \\ f_{a_{|\mathcal{A}|}} \end{pmatrix}$, the reward is identifiable up to constant by observing experts acting in $(T^1, \gamma_1), (T^2, \gamma_2)$ if and only if*

$$
rank \begin{pmatrix}
I - \gamma_1 T_{a_1}^1 & I - \gamma_2 T_{a_1}^2 & \mathbf{0} \\
\vdots & \vdots & \vdots \\
I - \gamma_1 T_{a_{|\mathcal{A}|}}^1 & I - \gamma_2 T_{a_{|\mathcal{A}|}}^2 & \mathbf{0} \\
I - \gamma_1 T_{a_1}^1 & \mathbf{0} & f_{a_1} \\
\vdots & \vdots & \vdots \\
I - \gamma_1 T_{a_{|\mathcal{A}|}}^1 & \mathbf{0} & f_{a_{|\mathcal{A}|}}
\end{pmatrix} = 2|\mathcal{S}| + d - 1. \tag{8}
$$

*where $f_a = (f_{s_1,a} \dots f_{s_{|\mathcal{S}|},a})^T \in \mathbb{R}^{|\mathcal{S}| \times d}$. On the other hand, if $\mathbf{1} \notin Im \begin{pmatrix} f_{a_1} \\ \vdots \\ f_{a_{|\mathcal{A}|}} \end{pmatrix}$, then the reward can be exactly recovered provided that the rank of the matrix on the left hand side of equation* (8), *which augments equation* (5) *by the features being matched, is $2|\mathcal{S}| + d$.*

Finally, it usually happens that the exact transition matrices $\{T_a\}_{a \in \mathcal{A}}$ are not known exactly and must be estimated, e.g., from samples. Verifying condition (5) on the approximated matrices may be misleading since the rank is very sensitive to small perturbations. Hence, we provide hereafter an identifiability condition in the case where we only have access to approximated transition matrices.

**Theorem 8.** *Suppose that we approximate the transition matrices $\{T_a^i\}_{a \in \mathcal{A}}$ as $\{\hat{T}_a^i\}_{a \in \mathcal{A}}$ such that $\|T_a^i - \hat{T}_a^i\|_2 \leq \epsilon \ \forall a \in \mathcal{A}, i = 1, 2$. Suppose that we verify condition (5) using the approximated matrices, i.e., we compute the second smallest eigenvalue $\sigma$ of the following matrix:*

$$
\begin{pmatrix}
I - \gamma_1 \hat{T}_{a_1}^1 & I - \gamma_2 \hat{T}_{a_1}^2 \\
\vdots & \vdots \\
I - \gamma_1 \hat{T}_{a_{|\mathcal{A}|}}^1 & I - \gamma_2 \hat{T}_{a_{|\mathcal{A}|}}^2
\end{pmatrix}. \tag{9}
$$

*Then, condition (5) on the true transition matrices $\{T_a\}_{a \in \mathcal{A}}$ holds provided that*

$$
\sigma > \epsilon \sqrt{2|\mathcal{A}|} \max(\gamma_1, \gamma_2). \tag{10}
$$

**Remark 2.** *The matrix estimator $\hat{T}_a$ can be obtained from samples. For example, [71][Lemma 5] shows that a high probability bound on the max norm $\|T_a - \hat{T}_a\|_{\max} \leq \epsilon$ requires $\mathcal{O}(\epsilon^{-4})$ samples from a generative model [72]. This would imply the following bound on the spectral norm: $\|T_a - \hat{T}_a\|_2 \leq |\mathcal{S}| \|T_a - \hat{T}_a\|_{\max} \leq |\mathcal{S}| \epsilon$. However, the dependence on $\epsilon$ can be improved as we show next applying the matrix Bernstein bound [73, 74].*

**Theorem 9.** *Let $\hat{T}_a$ be the empirical estimator for $T_a$. Then with probability greater than $1 - \delta$,*

$$
\|T_a - \hat{T}_a\|_2 \leq |\mathcal{S}| \sqrt{\frac{\log \frac{|\mathcal{S}||\mathcal{A}|}{\delta}}{2N}} + \frac{2(|\mathcal{S}| + 1) \log \frac{|\mathcal{S}||\mathcal{A}|}{\delta}}{3N} \quad \forall a \in \mathcal{A}. \tag{11}
$$

*Therefore, we can obtain $\|T_a - \hat{T}_a\|_2 \leq \epsilon$ with $\mathcal{O}(\epsilon^{-2})$ samples.*

## 4.2 Generalization to unknown environments

We now focus on reward generalizability, i.e., the ability to recover a reward function that would allow us to train an optimal policy in a new environment. Suppose that we recover a reward function that is compatible with two experts acting in two MDPs $\mathcal{M}_1, \mathcal{M}_2$, and that we use this reward to train an expert in a third environment $\mathcal{M}_3$, assuming all environments share the same true reward function but possibly different transition dynamics and discount factors. What condition guarantees that the trained expert will be optimal in $\mathcal{M}_3$?

This generalization requirement is milder than full reward identification. Indeed, being able to identify the reward (even up to a constant) naturally allows to train an optimal policy in any other environment sharing the same reward. However, even in the presence of non-trivial degrees of freedom, it may be the case that any recovered reward suffices to train an optimal policy in a given other environment.

Intuitively, the third training environment should not vary too much from the observed environments $\mathcal{M}_1, \mathcal{M}_2$. More precisely, if observing a third expert in environment 3 does not provide any further identification of the reward than with environments 1 and 2, then any reward compatible with environments 1 and 2 leads to an optimal policy in environment 3. The condition is made precise in the following theorem.

**Definition 10.** *Consider three Markov decision problems on the same set of states and actions, but with different transition matrices $T_1, T_2, T_3$ and discount factors $\gamma_1, \gamma_2, \gamma_3$. Suppose that we observe two optimal entropy regularized experts with respect to the same reward function in environments 1 and 2. We say that $(T^1, \gamma_1), (T^2, \gamma_2)$ **generalize to** $(T^3, \gamma_3)$ if any reward compatible with the two experts in environments 1 and 2 leads to an optimal expert in environment 3. The definition naturally extends to more than two observed experts.*

**Theorem 11.** $(T^1, \gamma_1), (T^2, \gamma_2)$ *generalize to* $(T^3, \gamma_3)$ *if and only if*

$$
rank\begin{pmatrix} I - \gamma_1 T_{a_1}^1 & I - \gamma_2 T_{a_1}^2 \\ \vdots & \vdots \\ I - \gamma_1 T_{a_{|\mathcal{A}|}}^1 & I - \gamma_2 T_{a_{|\mathcal{A}|}}^2 \end{pmatrix} = rank\begin{pmatrix} I - \gamma_1 T_{a_1}^1 & I - \gamma_2 T_{a_1}^2 & \mathbf{0} \\ \vdots & \vdots & \vdots \\ I - \gamma_1 T_{a_{|\mathcal{A}|}}^1 & I - \gamma_2 T_{a_{|\mathcal{A}|}}^2 & \mathbf{0} \\ I - \gamma_1 T_{a_1}^1 & \mathbf{0} & I - \gamma_3 T_{a_1}^3 \\ \vdots & \vdots & \vdots \\ I - \gamma_1 T_{a_{|\mathcal{A}|}}^1 & \mathbf{0} & I - \gamma_3 T_{a_{|\mathcal{A}|}}^3 \end{pmatrix} - |\mathcal{S}|.
$$

(12)

*This condition is also necessary, in the sense that, if it does not hold, then there exists a reward function compatible with experts 1 and 2 but which leads to a sub-optimal policy in environment 3.*

One interesting question is whether observing two experts in the same environment with different discount factors allows to generalize to any other expert with arbitrary discount factor. It turns out to be the case under some commutativity constraint on the transition matrices.

**Corollary 12.** *Consider a single environment with transitions $T$. Suppose that there exists an action $a_0 \in \mathcal{A}$ such that $T_{a_0}$ commutes with $T_a$ for all $a \in \mathcal{A}$. Then for any $0 < \gamma_1, \gamma_2, \gamma_3 < 1$ with $\gamma_1 \neq \gamma_2$, $(T, \gamma_1), (T, \gamma_2)$ generalize to $(T, \gamma_3)$.*

**Remark 3.** *The commutativity condition cannot simply be removed. Indeed, we provide in Appendix A.9 an example with two actions with non-commutative transition matrices for which condition (12) is not satisfied.*

## 5 Experiments

We now present empirical validations of our claims[1]. In particular, we verify the identifiability requirement given by Theorem 3 in the context of randomly generated transition matrices and different gridworlds with uniform additive noise in the dynamics.

In addition, we study a `Windy-Gridworld` and a financial model that we term `Strebulaev-Whited` both involving exogenous variables in their state spaces. In agreement with Corollary 6, the reward

---

[1]Code available at the following link `https://github.com/lviano/Identifiability_IRL`

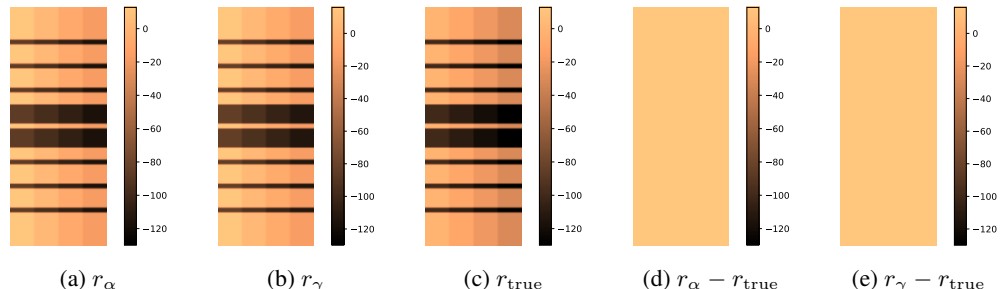

|      (a) $r_\alpha$      |      (b) $r_\gamma$      |      (c) $r_{\text{true}}$      |      (d) $r_\alpha - r_{\text{true}}$      |      (e) $r_\gamma - r_{\text{true}}$      |

Figure 1: Comparison between true and recovered reward in `Gridworld` with an action dependent reward, $|\mathcal{S}| = 100$. It can be noticed that the reward function $r_\gamma$ recovered changing discount factors is within a constant shift from the true reward ( subplots (b),(e)). The same conclusion holds for $r_\alpha$ recovered from different $\alpha$(see subplots (a),(d)).

function is not identifiable in these environments, highlighting the necessity of imposing milder requirements than full reward recovery. For example, in `Windy-Gridworld`, we show that by observing multiple experts acting in environments with different wind distributions, we can generalize, i.e., train an optimal expert in environments with arbitrary other wind distribution, in accordance with Theorem 11. On the other hand, in `Strebulaev-Whited`, given the additional information that the reward function can be represented as a linear combination of some known features, we can identify the reward, validating the condition of Theorem 7. The algorithms are described in Appendix B.

### 5.1 Identifiability experiments

**Experiments on `Random-Matrices`**   The first experiment involves randomly generated transition matrices and reward function with $|\mathcal{S}| = 18, |\mathcal{A}| = 5$. This setting matches the numerical evidence in [1]. Their algorithm recovers the reward function but the connection with their theoretical contribution is not highlighted. On the contrary, we have no theory practice mismatch, since we verify exactly the condition in Theorem 3. In particular, for the 100 random seed we tried the rank of the matrix $A$ is $2|\mathcal{S}| - 1 = 35$, then invoking Theorem 3 we can conclude that the reward function is identifiable up to a constant shift. We provide a visual example of the recovered reward in Figure 4 in Appendix C.

**Experiments on `Gridworld`**   As a second example of identifiability, we consider `Gridworld`, where the state space is a squared grid with 100 states while the action set is given by $\mathcal{A} = \{\text{up}, \text{down}, \text{left}, \text{right}\}$ with dynamics given by $T_\alpha(s'|s, a) = (1 - \alpha)T_{\text{det}}(s'|s, a) + \alpha U(s'|s, a)$ where $T_{\text{det}}(s'|s, a)$ represents deterministic transition dynamics where for example the action right leads to the state on the right with probability 1. If an action would lead outside the grid, then the agent stays in the current state with probability 1. The dynamics $U(s'|s, a)$ are instead uniform over the states that are first adjacent to the current state. In other words, $U(\cdot|s, a) = \text{Unif}(\mathcal{N}(s)) \quad \forall a \in \mathcal{A}$ where $\mathcal{N}(s)$ denotes the set of first neighbors of the state $s$.

We generate two different environments changing the value of $\alpha$, choosing $\alpha^1 = 0.4$ and $\alpha^2 = 0.2$. We notice that, even using the same discount factor $\gamma = 0.9$, the condition of Theorem 3 holds. When $\alpha$ is kept fixed, we also notice that the condition of Corollary 5 holds, and hence the reward can be recovered by just varying the discount factor $\gamma$ of the experts. We numerically verify that the reward can indeed be identified up to a constant shift in these two settings (see Figure 1).

### 5.2 Generalizability experiments

In this section, we present cases where identifiability is not possible due to the presence of exogenous variables. However, we notice that the generalizability condition in Theorem 11 is often satisfied, even for a test environment with parameters rather different than the environments of the observed experts. We start briefly describing the environments to later comment on the results.

**Experiments on `WindyGridworld`**   The `WindyGridworld` environment augments the `Gridworld` state representation by including a wind direction. The wind impacts the position transitions by making the agent move one step in the direction of the wind in addition to the action taken. The wind

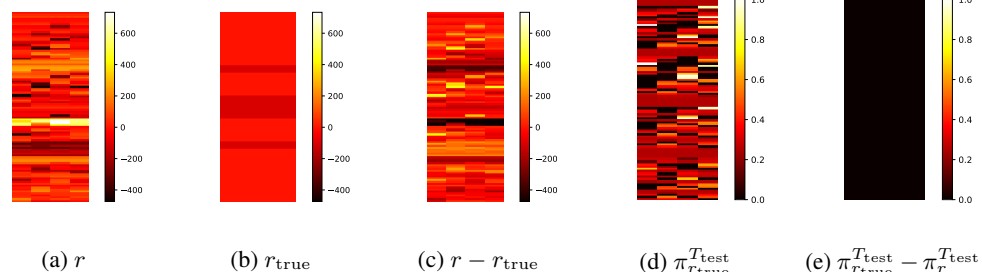

(a) $r$      (b) $r_{\text{true}}$      (c) $r - r_{\text{true}}$      (d) $\pi^{T_{\text{test}}}_{r_{\text{true}}}$      (e) $\pi^{T_{\text{test}}}_{r_{\text{true}}} - \pi^{T_{\text{test}}}_r$

Figure 3: Comparison between true and recovered reward ($r$ and $r_{\text{true}}$) from 4 experts in `WindyGridworld` with $|\mathcal{S}| = 400$. We notice that the reward function is not identified (see (a), (b), (c)). However, when we use the recovered reward in subplot (a) to train an optimal policy under unseen dynamics we recover the optimal policy under the true reward in subplot (b). The subplot (d) shows the policy $\pi^{T_{\text{test}}}_{r_{\text{true}}}$ recovered from the true reward in a new environment $T_{\text{test}}$ and (e) shows the difference between the policy recovered from $r_{\text{true}}$ and from the recovered reward denoted as $\pi^{T_{\text{test}}}_r$.

directions at step $t$, $w_t$ are sampled i.i.d. from the distribution $P_{\text{wind}}$, and is hence an exogenous variable. While the reward is not identifiable whatever the number of experts, we can generalize to a new environment with an arbitrary wind distribution by observing enough experts in environments with different wind distributions.

In Figure 2b, we see that we can obtain better identifiability (although never full identifiability) when increasing the number of experts. Once we have observed 4 experts, we do not get further identifiability by observing more experts, hence leading to generalizability as shown in Figure 2a and Figure 3. We conjecture that this number of experts is linked to the number of values that the exogenous variable, i.e. the wind direction, can take.

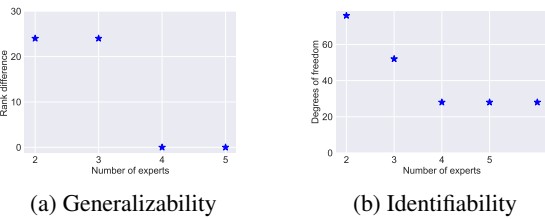

(a) Generalizability       (b) Identifiability

Figure 2: Figure 2a shows the difference between right and left term of Theorem 11. Figure 2b shows the difference between columns and rank of the matrix in Theorem 3. We have identifiability or generalizability respectively when those values are 0.

Furthermore, although the actions in Gridworld do not exactly commute (because of the boundary), observing two experts in the same environment with different discount factors enables generalizing to a different discount factor (see Figure 6 in Appendix C). The condition of Corollary 12 is hence sufficient but not necessary.

**Experiments on `Strebulaev-Whited`** The `Strebulaev-Whited` environment is the neoclassical investment model in which a firm has a Cobb-Douglas production function with decreasing returns to scale, as in [75]. The goal of the agent is to maximize profits discounted at rate $0 < \gamma < 1$. The state of the agent is defined by the capital level $k \geq 0$ and an exogenously given persistent stochastic productivity shock $z$. We can summarize the state by $s = (k, z)$. The next state $s' = (k', z')$ is determined separately for $k'$ and $z'$. We have that $k' = (1 - \delta)k + ak$, where $\delta$ is the depreciation rate of physical capital and $a$ is today's rate of investment which is the action in the model. The variable $z'$ evolves according to $\ln z' = \rho \ln z + \epsilon$ where $\epsilon \sim N(0, \sigma_\epsilon)$.

The continuous variables $k$ and $z$ are discretized according to the scheme proposed in [76]. Hence, we obtain a discrete process with $K^2$ possible values for the state variable $s = (k, z)$ (so $|\mathcal{S}| = K^2$) and $K$ values for the action $a$. In the experiments in Figure 7 in Appendix C , we choose $K = 20$ and consider two environments with different values of $\sigma_\epsilon$ set to $0.02$ and $0.04$, respectively. We observe that the rank of the identifiability matrix is 552. Since $552 < 2|\mathcal{S}| - 1 = 799$, the reward function is not identifiable up to a constant as expected in MDPs with exogenous states. Nonetheless, when we consider a third environment with $\sigma_\epsilon = 0.6$, the generalizability condition in Theorem 11 is satisfied. Hence, the expert behavior can be predicted in the third environment (see Figure 7e in Appendix C ).

## 5.3 Identifiability experiments with a restricted reward class

The final result presents a numerical validation of Theorem 7 in the environment `Strebulaev-Whited` with exogenous state variable. In this model, the true reward function can be expressed as a linear combination of the three features given by $f_{s,a} = [z((1-\delta)k+ak)^\theta, (1-\delta)k, ak]^T$, where $s = (k, z)$ and the parameter $\rho \in (0, 1)$ captures the curvature of the production function. We set $\rho = 0.55$. The first feature corresponds to the firm's output or sales which is available from the firm's income statement, the second feature is the firm's current capital stock net of depreciation which is available from the balance sheet, and the third feature is the level of investment that determines the future level of capital stock. The true reward function can be written as $r(s, a) = w^T f_{s,a}$ with $w = [1, 1, -1]^T$. It can be interpreted as follows: the agent's reward of investment is an increase in output/sales, $w_1 = 1 > 0$, while the cost of capital is 1 and, hence, investment is costly, $w_3 = -1 < 0$. At the same time, the capital stock is valuable and can be liquidated at a price of $w_2 = 1 > 0$.

Knowing these features, we can verify that the rank of the matrix in Equation (8), is 803 which is equal to $2|S| + d$ in this environment ($|S| = 400$ and $d = 3$). Invoking Theorem 7, we can conclude that the reward function is identifiable exactly, which is verified numerically in Figure 8 in Appendix C. Expressing the reward in terms of features hence helps identifiability and interpretability.

## 6 Conclusion

In this paper, we analyze conditions that guarantee identifiability of the reward function (up to an additive constant) from multiple observed experts maximizing the same reward and facing different transition dynamics. This allows us to train optimal policies in any other environment sharing the same reward with the environments of the observed experts. On the other hand, in order to generalize to unknown environments, such strong reward identification is not required, and we provide a milder necessary and sufficient condition for generalizability. We also provide identifiability results in a variety of settings, i.e., linearly parameterized reward, approximated transition matrices, observation of any number of experts, as well as a non-identifiability result in the presence of exogenous variables. In the following, we list the main limitations of our work that will be the subject of future studies.

**Observing experts in different environments.** We saw that observing a single expert in one environment cannot lead to reward identification in our setting. We hence need to observe at least two experts acting in different enough environments. To motivate this assumption, note that varying environments are ubiquitous in RL, in particular in Robust RL which deals with the training of experts that perform well in different environments, where the transition dynamics can vary to some extent. It is hence rather common to consider that the transition dynamics of a given environment can change. This was studied, e.g., in [77–80], where the authors considered different Mujoco environments with varying friction coefficients, or object masses, which influence the dynamics. Also, instead of observing different experts in different environments, we could imagine that we observe a single expert in a single environment that varies over time (but with fixed reward), and that the expert adapts to these changes. Such observations would provide us optimal actions in environments with different transition dynamics, and thus our results would apply. This is of particular interest in economics/finance where the environment is in constant evolution.

**Assuming entropy regularized experts.** When observing real world data, we have to face the fact that humans do not follow this idealized mathematical model. However, it turns out that our results still hold for the more general class of regularized MDPs [81] where we replace the entropy with any strongly convex function (see Appendix D). Whether the flexibility in the choice of the strongly convex regularizer allows to better capture real-world behaviors is an open question.

## Acknowledgements

This work has received financial support from the Enterprise for Society Center (E4S) and SNF project 100018_192584. This work was supported by the Swiss National Science Foundation (SNSF) under grant number 200021_205011. This project has received funding from the European Research Council (ERC) under the European Union's Horizon 2020 research and innovation programme (grant agreement n° 725594 - time-data).

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
