# OpenReview forum: "Identifiability and generalizability from multiple experts in Inverse Reinforcement Learning"
_NeurIPS.cc/2022/Conference — NeurIPS 2022 Accept_

### Official Review · Reviewer_P6wR · 2022-07-04

**Rating:** 6
**Confidence:** 3
**Soundness:** 3 good
**Presentation:** 3 good
**Contribution:** 2 fair

**Summary:**

The authors propose an approach to solving the ill-posedness of IRL by exploiting multiple experts. In particular, a necessary and sufficient condition on the expert environments is derived, and further extended in different scenarios (e.g. approximated dynamics, linearly parametrized reward). Besides a negative result, a milder generalization requirement is given such that any recovered reward suffices to train an optimal policy in a given other environment.

The work is mainly theoretical (even though the conditions are numerically verifiable), and limited by a number of strong requirements, but the results are clearly stated, their impact is well described, and the findings constitute a good contribution to the literature.

**Questions:**

1) In Theorem 7, does the real reward need to belong to the same parametrized space of the recovered reward? Please clarify this in the text

2) The work offers only theoretical conditions to check if the expert's reward con be properly reconstructed but does not comment on how this can be done. Which IRL algorithm can be used to be sure that the recovered reward is actually compatible with the expert's demonstrations?
For example, in the numerical results, how was the reward identified?

**Strengths And Weaknesses:**

The paper is well written, and it is mathematically sound. The theoretical results are well described, their requirements clearly stated, and their impact properly discussed.

---

> ### Author Response · Authors · 2022-07-29
> **Answers to Reviewer P6wR**
>
> **“In Theorem 7, does the real reward need to belong to the same parametrized space of the recovered reward? Please clarify this in the text”:**
>
> Yes, the true reward should be in the considered set of restricted rewards. This is indeed not clear from the current theorem statement, we will modify it accordingly.
>
> **About the reward recovery algorithm:**
>
> The ways the rewards are recovered in the experiments are described in Algorithms 1, 2 and 3 in Appendix B. The difficulty in the current setup is not to find a compatible reward with the observed experts (this can simply be done by finding a solution to a linear system), but rather to know whether the recovered reward matches the ‘true’ reward up to an additive constant.

---

### Official Review · Reviewer_Pbuu · 2022-07-10

**Rating:** 6
**Confidence:** 4
**Soundness:** 4 excellent
**Presentation:** 4 excellent
**Contribution:** 2 fair

**Summary:**

This paper investigates the problem of identifying a reward function (up to a constant) having access to (entropy-regularized) optimal policies of multiple experts on different environments and with different discount factors (this result was previously known only with two experts). In addition to this theoretical result, the authors provide a practical method for identifying such a reward. Finally, when assuming the reward is a linear combination of features, the authors demonstrate an interpretable reward function can be identified.
The authors present some empirical results on random matrices and grid worlds that showcase and confirm their theoretical results.

**Questions:**

1. A high-level question I had is: wouldn't it be more realistic to assume that you have multiple experts operating with different _rewards_ as opposed to different environments/discounts? I realize this is a completely different problem, but I feel it should at least be discussed when motivating the problem (see points on significance above).
2. In line 157 it says that the condition of Theorem 3 "is easier to check in practice" than Definition 2, why is this? Is it because there's no need of the $v$s? It would be good to clarify.
3. In the proof of Theorem 3 the statement "is an eigenvector of A with eigenvalue 0, and corresponds to the invariance of the optimal policy under addition of a constant to the reward function." is not clear).
4. The statement in line 586 in Appendix ("This is hence equivalent...") is not clear at all. Please add some more clarifying text or a reference to an established result (it's fine if it points to a chapter in a textbook, say).

In general, statements of the form above (which are not trivially true) should be clarified or referenced rather than left "as an exercise to the reader".

5.  In line 313, is $\beta$ essentially just the standard $\gamma$ discount factor?

**Limitations:**

I don't feel the authors added much discussion around limitations of their work; most of the discussions and results served only to highlight how their results improved on prior results. From a practical perspective it would have been nice to have a bit more of a nuanced discussion here (see points above regarding significance).

No negative societal impact of the work was provided.

**Strengths And Weaknesses:**

# Originality
The work seems to be largely a followup of [1], but as far as I can tell, the work is original (including theory and experiments). I went through the proofs and as far as I can tell they appear to be novel and non-trivial (e.g. not just a direct application of prior results).

# Quality
The theoretical results and proofs are of high quality and clarity. The empirical validations were done very well and properly highlighted and confirmed the theoretical results of the paper.

# Clarity
The paper was in general very well written and presented. The motivating paragraphs before each Theorem were very clear and useful in giving a high-level intuition for the results.
A few suggestions to help make the paper clearer:
* In Definition 2 I'd suggest using subscripts (instead of superscripts) for the $v^1$, $v^2$s and $T^1$, $T^2$s, as otherwise they can be confused with exponents.
*  The paragraph describing the Strebulaev-Whited environment is quite difficult to follow. Although I appreciate the authors want to add a real-world-like environment, I feel it is mostly a distraction from the paper's flow and fails to effect its intended purpose (due to lack of clarity). Indeed, the actual figures are relegated to the appendix, so the main paper only contains a (too terse) description of this non-standard environment. I would suggest moving the environment's description to the appendix as well.

# Significance
This is probably the weakest point of the paper for me. Although the authors present some high-level motivations in the first paragraph of the introduction, and then position it as resolving some of the issues with IRL, it remained unclear to me _when_ such a situation as explored by the authors would be useful. The Strebulaev-Whited environment, although less toy-ish than the grid worlds, is still quite artificial and unclear whether the situation considered (with different values of $\sigma_{\epsilon}$ has any ties to any real world situation.

These issues could have perhaps been addressed in a concluding section, but the authors chose not to include one. I would suggest removing the Strebulaev-Whited environment description (as suggested above) and adding a conclusion instead. In particular, I would suggest they clarify _when_ these methods can be useful and ideally give examples of situations where you have multiple experts operating under different environments/discount factors but with the same reward function.

---

> ### Author Response · Authors · 2022-07-29
> **Answers to Reviewer Pbuu**
>
> **About observing experts in the same environment w.r.t. different reward functions:**
>
> Identifying independent rewards of two experts, even acting in the same environment, is a separable problem, in the sense that one expert will not provide any information on the reward of the other expert. Hence, the problem would remain ill-defined.
>
> The second expert should provide additional information about the reward of the first expert. It seems that assuming the same reward for the second expert but different environment indeed provides new information. It is an interesting question whether we can relax a bit the ‘equal rewards’ assumption, and only account for ‘similar rewards’. This is in some way already the case in this work since we assume possibly different discount factors. We believe that our results provide useful tools to extend IRL to other setups.
>
> **About the hardness of checking the condition in Cao et al.:**
>
> The value distinguishing assumption of Cao et al. states that there exist no pair of non-constant vectors $(w, w’)$ such that a certain equality holds, which includes $w,w’$ and the transition matrices of the two environments (equation (7) in Cao et al.). Naively checking this assumption would hence require testing this statement for any pair $(w,w’)$ of non-constant vectors, which is computationally impossible. In contrast, our restatement allows us to check the identifiability condition in finite time.
>
> **"is an eigenvector of A with eigenvalue 0, and corresponds to the invariance of the optimal policy under addition of a constant to the reward function." is not clear:**
>
> This comment is meant to explain where the rank deficiency comes from. We see that the eigenvector of A  with eigenvalue 0 is formed of two constant vectors associated with each value vector of both experts. This means that adding a constant to the value vector of both experts would still provide a solution to the linear system (14). Moreover, looking at the reward decomposition in terms of value function (below line 555), we see that adding a constant to the value vector also yields the addition of a constant to the reward.
>
> **“The statement in line 586 in Appendix ("This is hence equivalent...") is not clear at all:**
>
> First, note that the vector pair $(v^1, v^2)$ on the right of the equation below line 585 is always a solution of the linear system on the left. Hence, we can trivially replace the implication sign ($\Rightarrow$) in this line with an equivalence sign ($\Leftrightarrow$). Therefore, the condition of Definition 2 precisely describes the solutions of the linear system on the left, i.e., the kernel of the matrix written below line 586. The condition states that this kernel must be formed by vector pairs of the form $(v^1, v^2) = (c\mathbf{1}, c(1-\gamma_1)/(1-\gamma_2) \mathbf{1})$ for $c \in \mathbb{R}$, which is a vector space of dimension 1.
>
> **“In line 313, is β essentially just the standard γ discount factor?”:**
>
> Yes, β is the standard discount factor. We will replace the notation.
>
> **Significance**
>
> Please refer to the common response.
>
> We thank the reviewer for the suggestions on improving the clarity. These will be taken into account in the final version.

---

> > ### Comment · Reviewer_Pbuu · 2022-08-04
> > **Thanks**
> >
> > Thank you for your response to my review, as well as the common response. They have mostly addressed my concerns. I look forward to reading an updated version of the paper with an added conclusion, as this will hopefully help address the concerns around significance.

---

> > > ### Author Response · Authors · 2022-08-07
> > > **Added conclusion**
> > >
> > > Thank you for your response.
> > >
> > > We have just posted a conclusion draft in the common thread. We hope this will help to address the significance concerns. Moreover, we welcome any further feedback from your side.

---

### Official Review · Reviewer_4M5d · 2022-07-10

**Rating:** 7
**Confidence:** 4
**Soundness:** 3 good
**Presentation:** 3 good
**Contribution:** 2 fair

**Summary:**

This IRL theory paper presents a new statement of an identifiability result (first shown in [1]) for recovering a reward function given observations of multiple experts with different discount factors and/or acting under different transition dynamics in tabular MDPs. This paper improves the previous result from [1] by stating the main indentifiability result as a more practically useful matrix rank condition which can be easily verified (Thm. 3). This allows the authors to extend the result to several interesting cases - such as more than 2 experts (Corr. 4), MDPs with 'exogenous variables' (Corr. 6), Linear reward features (Thm. 7), and sample estimated transition dynamics matrices (Thm. 8). The authors also show how their result can be used to derive a reward generalization result (Thm. 11) under weaker conditions, and present numerical experiments on synthetic tabular MDPs to verify their theoretical results.

I have checked the in-text derivations fairly closely and am familiar with prior theory in this area. I have not checked the proofs or extended derivations in the appendices.

**Questions:**

 * The generalization pattern from 2 experts to 3 (and beyond) is not clear to my by comparing Eq 5. and Eq. 6. I.e. I would not be able to predict the structure of the equivalent rank condition for 4 experts. Can you please clarify exactly how the rank condition generalizes as the number of experts increases beyond 3? This is particularly important as other results (Thm. 7, 11) build on this expression.

 * Line 28 - 'highly parameterized and represented by a low-dimensional set of parameters' is a little unclear - are you alluding to Deep Neural Network reward representations here?

 * Line 121 - the main advance beyond [1] is that the identifiability result is easy to verify in practice (as stated on Line 121). Can you please include a brief sketch / outline in your paper illustrating why the result in [1] is _not_ easy to verify in practice, for readers not familiar with this work.

 * Line 243 - "$T_{a_0}$ commutes with $T_a ~~\forall a \in \mathcal{A}$". It is difficult to intuit / conceptualize what this condition implies for the structure of the MDPs. Can you please provide a brief explanation of what this commutativity requirement implies for the MDP dynamics? Perhaps with a small example?

**Limitations:**

Nil comments here.

**Strengths And Weaknesses:**

# Originality

Although this paper is a re-statement of an existing result from [1] (as the author's acknowledge), this re-statement is more practically useful and enables multiple extensions, as well as numerical validation and application of the result.

# Quality

The authors have done a good job engaging with prior work. Overall, this paper is of high quality and feels solid and complete, although the paper is presently missing a conclusion section / paragraph - I encourage the authors to add one to their final submission.

# Clarity

This paper is clearly written, although some of the notation, language and definitions could be tidied up. For instance;

 * The language switches between "Markov Decision Process' (ex. line 91) and '... Problem' (ex. Line 117). Please be consistent unless these imply different things in your writing, in which case both terms should be defined.
 * Line 92 - it may clarify the definitions to explicitly state that $\mathcal{S}$ and $\mathcal{A}$ are finite.
 * Line 115 - Do you mean '[additive] constant'?
 * On line 119 and elsewhere $v$ denotes a _function_, whereas on line 137 and following it now denotes a _vector_. Please clarify the notation to disambiguate these if possible.
 * Consider annotating matrix expressions (e.g. Eq. 4) with the dimensions of each term, to help the reader understand the notation. Alternately, I suggest adding a 'bar' between the columns of block matrices such as the leftmost term in Eq. 4 to clarify the matrix contents.

The inclusion of an 'intuition' for the proof of Def. 10 (Lines 225-229) is very helpful. Please consider adding similar intuition / proof sketch text for more of your other results. This will aid less-technical readers in following the material.

Please ensure figures are legible in greyscale. E.g. consider using a log color scale for Fig 1d and 1e, as well as Fig 3e. Please include x and y axis labels for Figs 1 and 3 (I believe the axes are states and actions?).

# Significance

This paper contributes a well scoped and executed body of theory work to the IRL literature, and creates multiple opportunities for future authors to build on the results. For instance, by extending the tabular MDP results to derive probabilistic bounds in approximate IRL settings, or by applying the results in practical applications of IRL to real-world MDP problems.

---

> ### Author Response · Authors · 2022-07-29
> **Answers to Reviewer 4M5d**
>
> **About generalizing to any number of experts**
>
> The generalization to any given number of experts involves the matrix $A_n$ given in equation (45) (the ‘-’ signs can be removed since we will only compute the rank of the matrix), and the identifiability condition reads $\text{rank}(A_n) = n|S| - 1$. We only wrote and proved the results for the case of 3 experts since it was particularly relevant for stating the generalizability results. The proof for n experts follows the same line, but we will add it in the appendix of the final version.
>
> **Line 28 - 'highly parameterized and represented by a low-dimensional set of parameters' is a little unclear - are you alluding to Deep Neural Network reward representations here?**
>
> It is a bit the other way around. We mean that in some applications, the reward is parameterized only using a small set of parameters. In the economics/finance literature, the reward function is often represented by simple concave utility functions, such as quadratic, exponential, or power functions known as CARA, CRRA, or HARA utility. In addition, the reward function is often assumed to have a single argument, e.g., consumption. For instance, the classic Mehra-Prescott equity premium puzzle [1] has been established using the constant relative risk aversion utility, $r(a)=(a^{1-\alpha}-1)/(1-\alpha)$, and the single parameter estimated in applied work (e.g., [2]) is the relative risk aversion coefficient \alpha.
>
> While only considering a small family of possible rewards leads to better identification, this may restrict its range of validity, and in particular its generalization to tasks with different transition dynamics. In this work, we consider all possible state/action dependent reward functions.
>
> We will rephrase this part of the introduction.
>
> [1] Rajnish Mehra, Edward C. Prescott: The Equity Premium: A Puzzle. In: Journal of Monetary Economics. Nr. 15, 1985, S. 145–161
>
> [2] Hansen, Lars Peter, and Kenneth J. Singleton. 1982. “Generalized Instrumental Variables Estimation of Nonlinear Rational Expectations Models.” Econometrica 50 (5): 1269–86.
>
> **About the hardness of checking the condition in Cao et al.:**
>
> The value distinguishing assumption in Cao et al. states that there exist no pair of non-constant vectors $(w, w’)$ such that a certain equality holds, which includes $w,w’$ and the transition matrices of the two environments (equation (7) in Cao et al.). Naively checking this assumption would hence require testing this statement for any pair $(w,w’)$ of non-constant vectors, which is computationally impossible. In contrast, our restatement allows to check the identifiability condition in finite time.
>
> **About the matrices commutation condition:**
>
> $T_{a_0}$ commutes with $T_a$ means that, in the MDP, taking first action $a_0$ and then action $a$ leads the agent to the same distribution over state as if it had taken $a$ first and then $a_0$. For deterministic dynamics this means that the final state does not depend on the order of the actions (although the obtained reward can still be different!). In gridworld, in general, going $up$ and then $right$ leads to the same state as going $right$ and then $up$ (they both lead to the state on the upper right of the current state). This property only fails when the agent is near the edge of the grid. For example, if there is a wall on the right, taking $left$ and then $right$ leads the agent to the same original state, but taking $right$ and then $left$ leads it to one state on the left, since the first $right$ action was blocked by the wall.
>
> However, it seems that these small contradictions to the commutation assumption did not affect identifiability in the case of gridworld, as shown in the experiments.
>
> **Line 115 - Do you mean '[additive] constant'?**
>
> Yes.
>
> We thank the reviewer for the suggestions on improving the clarity. These will be taken into account in the final version.

---

> > ### Comment · Reviewer_4M5d · 2022-08-02
> > **Resposne to author response**
> >
> > Thank you for your responses to my queries - this all makes sense. In particular, your example to explain the matrix communtation condition is really insightful and I would encourage you to include this in the paper or appendix! Great work with this paper - I enjoyed reading it :)

---

### Official Review · Reviewer_z9k4 · 2022-07-11

**Rating:** 4
**Confidence:** 3
**Soundness:** 3 good
**Presentation:** 2 fair
**Contribution:** 2 fair

**Summary:**

Inverse Reinforcement Learning (IRL) seeks to infer a reward function from demonstrations sampled from an agent acting (approximately) optimally. Unfortunately, the IRL problem is under-determined: an infinite number of reward functions will lead to exactly the same optimal policy. However, it is possible to obtain additional information when observing demonstrations from multiple experts, acting in environments that differ in transition dynamics or discount factor (but which have the same state and action space, and reward function).

This paper characterizes when this multi-expert IRL problem is or is not identifiable. The characterizations are in terms of the rank of matrices formed from the transition matrices of the different environments. In particular, Theorem 3 provides a charcterization in the two-expert, distinct dynamics case that is equivalent to that of Cao et al [1] but which is more amenable to direct validation. They derive corollaries for the three-expert case (Corollary 4) and two different discount rate (Corollary 5). There is also a key negative result (Corollary 6), showing that when there are exogeneous variables then the reward function can only be determined up to an additive function of the exogeneous variable (not a constant). They also derive stronger results in the case of a linear reward function (Theorem 7), and provide a condition suitable for estimated transition matrices (Theorem 8).

The paper concludes with Theorem 11 providing a condition for a reward function to generalize. This and other theorems are then illustrated in a brief empirical section.

 Some of these transformations -- such as potential shaping, or rescaling by a positive constant -- we might

**Questions:**

  - Most of the results in the paper describe whether or not IRL can characterize the reward function up to a constant. But why should we care about this? Isn't it sufficient to identify it up to potential shaping, since that never changes the optimal policy or ranking of policies?
  - How might these results change if expert is not MaxEnt RL? E.g. no entropy bonus (hard optimal policy), or a different kind of suboptimality (epsilon-greedy)?

**Limitations:**

I did not see any substantive discussion of limitations of this work. One obvious limitation is it assumes a particular model of the demonstrator (MaxEnt RL). Humans, unfortunately, do not follow this idealized mathematical model. This is a limitation common with much work in this area -- but it'd be nice to at least see some discussion of how the results might vary with different demonstrator models, to get a sense of how robust these results are.

No discussion of ethical implications but I also don't see any significant negative ethical implications from this work.

**Strengths And Weaknesses:**

Originality: moderate. The key result, Theorem 3, is equivalent to an earlier result due to Cao et al. Most of the other results are relatively simple applications of this theorem. However, Theorem 3 is a non-trivial restatement, and the corollaries although not technically deep are a useful set of results.

Quality: all the claims seem plausible although I have not had time to validate the proofs in detail.

Clarity: comprehensible but significant room for improvement. The actual technical description was fairly clear. But there is little to signpost the reader in the introduction, and I even found the introduction misleading in places, e.g. you refer to "reward shaping" but later it is clear you are talking specifically about *potential shaping* (but for some reason do not cite the Ng et al, 1999 paper until later).

The motivation in the introduction focusing on economics & finance also felt rather odd. With the benefit of hindsight, I suppose this is meant to foreshadow the empirical results, and perhaps rewards linear in features are also more plausible in this setting. But in general, I do not think of this as being a major application of IRL. Indeed, it seems relatively easy to hand-design a passable reward function for many financial applications, e.g. PnL minus lambda*CVAR. By contrast, it seems far more challenging to work in settings that do not have directly measurable metrics of success, such as "help the user build a house in Minecraft" (BASALT benchmark, etc).

Moreover, citations [5]-[8] seem only tangentially related (prospect theory, VnM utility theorem, etc). At best they establish the "fundamental importance" of reward functions. But not any difficulty in specifying them.

Paper would also benefit from having a conclusion section, and limitations and future work.

Significance: moderate. The matrix rank conditions are more amenable to empirical validation than prior work -- especially given the approximation condition of Theorem 8. The results for generalization are also a useful application, and notably can give guarantees of generalization even when the reward is not perfectly identifiable (depending on the environment to be transferred to).

Minor typos/grammar points I'd suggest fixing:
  - "However, [1] showed that" -- [1] is not a proper noun, should read "However, Cao et al [1] showed that". (This misuse occurs many times later on in related work, as well.)
  - "This work starts by" -- is this work [1], or your paper?
  - "maximized by the agent" -- agent is ambiguous, could refer to the policy you learn, call it demonstrator instead?
  - Line 94: R should be r (lowercase) for consistency with line 91 and 97.
  - References: some words in titles need to be capitalized, plae inside {...} in the TeX, e.g. monte carlo->Monte Carlo.

# Update after rebuttal

Thanks to the author for their detailed response. I understand the Econ motivation better now, IRL is certainly relevant for understanding human behaviour in financial transactions. I still find spotlighting this application a bit strange given that as the authors say themselves "IRL has not (yet) been used in revealed preference analysis". It'd be interesting to see work applying IRL to human financial decisions and see if it's more predictive than existing Econ models --- but this paper does not do that, so why focus on it?

I'm still not sure in what setting we'd care about reward functions up to a constant rather than up to potential shaping as we know potential shaping does not change the ranking of policies. The author response doesn't really address this point.

---

> ### Author Response · Authors · 2022-07-29
> **Answers to Reviewer z9k4**
>
> **About the confusion between "reward shaping" and "potential shaping":**
>
> Given the Remark 3 in Cao et al., 2021 we know that the set of rewards that leaves the optimal policy unchanged is completely characterized by the potential shaping transformation presented in Ng et al., 1999. At this point, the two terminologies may be used interchangeably. Please let us know if we are missing something and we should use one terminology rather than the other.
>
> **About the application of IRL to finance:**
>
> The economics/finance literature differentiates between axiomatic and revealed preference theory. In axiomatic preference theory, the utility/reward function is posited or derived from basic axioms. Here, indeed, there is no difficulty in specifying a reward function, such as PnL minus lambda*CVAR. In empirical and experimental work, however, simple reward function specifications are often rejected and consumers/investors have been shown to exhibit behavioral biases and/or non-standard preferences.
>
> This work relates more to the vast literature on revealed preference, which we should have highlighted more and differentiated better. Revealed preference theory, initiated by [1, 2], provides an approach to analyze actions (e.g., consumer’s demand behavior) by assuming that observed choices provide information about the underlying preferences, or reward function. Revealed preference theory is, hence, similar in spirit to IRL. But IRL has not (yet) been used in revealed preference analysis. [3, 4] provide excellent reviews of recent advances in revealed preference theory.
>
> The goal of revealed preference theory is to recover the agents’ preferences. This task is important because knowledge of the reward function is required to conduct counterfactual policy analysis. Knowing the policy function is not enough. In financial applications, for instance, the impact of a Tobin tax can be assessed only knowing investors’ preferences for trading (see, e.g., [5]).
>
>
> [1] Samuelson, P.A. (1938) “A Note on the Pure Theory of Consumer’s Behavior,” Economica, 5: 61-71.
>
> [2] Samuelson, P.A. (1948) “Consumption Theory in Terms of Revealed Preference,” Economica, 15: 243-253.
>
> [3] Demuynck, T., Hjertstrand, P. (2019). Samuelson’s Approach to Revealed Preference Theory: Some Recent Advances. In: Cord, R., Anderson, R., Barnett, W. (eds) Paul Samuelson. Remaking Economics: Eminent Post-War Economists. Palgrave Macmillan, London.
>
> [4] Echenique, F., (2019) “New developments in revealed preference theory: decisions under risk, uncertainty, and intertemporal choice”, arXiv e-prints.
>
> [5] Tobin, J., (1978), “A Proposal for International Monetary Reform,” Eastern Economic Journal, Vol. 4 (July-October), pp. 153–59.
>
>
> **About motivations for reward identification and the use of entropy regularized experts:**
>
> See the common response.
>
> We thank the reviewer for the spotted typos which will be fixed.

---

> > ### Comment · Reviewer_z9k4 · 2022-08-06
> > **Clarification on reward shaping**
> >
> > > Given the Remark 3 in Cao et al., 2021 we know that the set of rewards that leaves the optimal policy unchanged is completely characterized by the potential shaping transformation presented in Ng et al., 1999. At this point, the two terminologies may be used interchangeably. Please let us know if we are missing something and we should use one terminology rather than the other.
> >
> > Reward shaping is frequently used to refer to heuristic terms added to aid policy exploration. These may change the optimal policy, though are often annealed to zero as training progresses. https://link.springer.com/referenceworkentry/10.1007/978-0-387-30164-8_731 gives several examples of this more general usage.

---

> > > ### Author Response · Authors · 2022-08-07
> > > **Further clarification of our motivation, and the relation to reward shaping**
> > >
> > > Let us clarify the connection between our work and potential shaping. Potential shaping describes the degrees of freedom in the reward that leave the optimal policy (or ranking of the policies) invariant within a certain environment. The existence of such degrees of freedom actually implies the non-identifiability of the reward function from a single expert, and is fully described in Theorem 1.
> > >
> > > However, while there always exists a certain flexibility of the reward in any given environment, the space of rewards that leaves the optimal policy invariant varies in different environments. This can again be seen from Theorem 1 where we observe that the degree of freedom in $r$, characterized by the choice of value function $v$, depends on the transition matrices $T$.
> > >
> > > The main implication of this observation is that knowing the reward function up to potential shaping in a given environment does **not** allow to train an optimal policy in a different environment.
> > >
> > > In order to generalize to **any other environment**, identification of the reward up to an additive constant is **necessary**. On the other hand, if one wants to only generalize to a given environment, such strong reward identification is not required, and Theorem 11 provides a milder necessary and sufficient condition for it. This necessity is made clear in the Theorem statement: “if it does not hold, then there exists a reward function compatible with experts 1 and 2 which leads to a sub-optimal policy in environment 3”.

---

> > > > ### Comment · Reviewer_z9k4 · 2022-08-07
> > > > **Querying potential shaping connection**
> > > >
> > > > > The main implication of this observation is that knowing the reward function up to potential shaping in a given environment does not allow to train an optimal policy in a different environment.
> > > >
> > > > Doesn't this claim directly contradict Theorem 1 of [Ng et al (1999)](https://people.eecs.berkeley.edu/~pabbeel/cs287-fa09/readings/NgHaradaRussell-shaping-ICML1999.pdf) which shows that the optimal policy is identical in reward functions related up to potential shaping for any MDP?
> > > >
> > > > I think you get an even broader class of invariances if only targeting a specific environment.

---

> > > > > ### Author Response · Authors · 2022-08-07
> > > > > **We think that there is no contradiction with Ng et al., 1999**
> > > > >
> > > > > Thank you for your question.
> > > > >
> > > > > We do not think there is a contradiction between our statement and Theorem 1 in Ng et al. 1999.
> > > > >
> > > > > Indeed, using Ng et al., 1999 notation, their Theorem 1 states that an optimal policy in the MDP $ M = (S,A,T, \gamma, R + F)$ is optimal also in $ M^\prime = (S,A,T, \gamma, R)$. Notice that this statement holds because both MDPs $M$ and $M^\prime$ share the same transition dynamics $T$.
> > > > >
> > > > > Our setting is different because we consider that the two environment may have different transition dynamics, i.e. $ M = (S,A,T, \gamma, R + F)$ and $ M^\prime = (S,A,T^\prime, \gamma, R )$. In this case, Theorem 1 in Ng et al. 1999 does not apply, that is, an optimal policy in $M$ is not necessarly optimal in $M^\prime$.

---

> > > > > > ### Author Response · Authors · 2022-08-09
> > > > > > **Message to Reviewer z9k4**
> > > > > >
> > > > > > Dear reviewer,
> > > > > >
> > > > > > as the discussion period ends soon we were wondering if you have any other question on the differences between our work and (Ng et al. 1999).  If that is the case, we are happy to have further discussion.
> > > > > >
> > > > > > Best,
> > > > > > Authors

---

### Author Response · Authors · 2022-07-29
**Common response**

We thank the reviewers for their valuable and globally positive comments.

The main concern, raised by two of the reviewers (z9k4 and Pbuu), is the significance of the studied setup, i.e., reward identification from observing entropy regularized experts acting in different environments. More precisely, the three raises concerns were: The purpose of reward identification up to a constant rather than up to potential shaping (Reviewer z9k4), the plausibility of observing agents acting in different environments with respect to the same reward function (Reviewer Pbuu), and the soundness of considering entropy regularized experts as opposed to other types of optimal experts (Reviewer z9k4). We hence provide additional motivations for these 3 aspects.

**Reward identification:**

As mentioned in the introduction in [4],
“in many applications it is not enough to find some pattern of rewards corresponding to observed policies; instead we may need to identify the specific rewards agents face, as it is only with this information that we can make valid predictions for their actions in a changed environment. In other words, we do not simply wish to learn a reward which allows us to imitate agents in the current environment, but which allows us to predict their actions in other settings.”

In some cases, we may however not need to recover the exact reward in order to generalize to another given environment. This is why we also consider the more restricted goal of generalization instead of reward identification.

**Observing experts from different environments:**

Varying environments are ubiquitous in RL, in particular in Robust RL which deals with the training of experts that perform well in different environments, where the transition dynamics can vary to some extent. It is hence rather common to consider that the transition dynamics of a given environment can change. This was studied, e.g., in [1,2,3], where the authors considered different Mujoco environments with varying friction coefficients, or object masses, which influence the dynamics.

Also, instead of observing different experts in different environments, we could imagine that we observe a single expert in a single environment that varies over time (but with fixed reward), and that the expert adapts to these changes. Such observations would provide us optimal actions in environments with different transition dynamics, and thus our results would apply. This is of particular interest in economics/finance where the environment is in constant evolution.

**Entropy regularized experts:**

It turns out that our identifiability result is valid more generally for regularized MDPs [5] where the entropy term in equation (1) is replaced by any other strongly convex differentiable function of the policy $\Omega(\pi)$.

Indeed, we can use Proposition 1 and Definition 1 in [5] to establish that for any value vector $v$ and reward $f$, there exists a unique policy that satisfies $$\pi(a|s) = \nabla \Omega^\ast( f(s,a) + \gamma \sum_{s^\prime}P(s^\prime|s,a)v(s^\prime)).$$

where $^\ast$ denotes the Fenchel conjugate. By the distributivity property (iii) in Proposition 1 of [5], we can subtract a function dependent only on state in the argument without affecting the equality. This gives that for any $v$ and $f$, there exists a unique $\pi$ such that
$$\pi(a|s) = \nabla \Omega^\ast( f(s,a) + \gamma \sum_{s^\prime}P(s^\prime|s,a)v(s^\prime) - v(s))$$

Using the convexity of $\Omega$, we have that $\nabla \Omega$ is the inverse map of $\nabla \Omega^\ast$. Hence we obtain
$$\nabla \Omega(\pi(a|s)) = f(s,a) + \gamma \sum_{s^\prime}P(s^\prime|s,a)v(s^\prime) - v(s)$$
which is  the equivalent of our Theorem 1 for general strongly convex regularizers. The only part changing is the left hand side. However, we saw in the analysis that reward identifiability was not depending on this part of the equation. When using a different regularizer, the recovered reward given observed expert policies will be different, but the identifiability condition remains the same.

However, epsilon-greedy or deterministic greedy policies would not fit this setting. Identifiability is more challenging with these kinds of experts because the knowledge of such policies only informs us with the action yielding the highest expected value, but no information about the relative difference with respect to other actions, in contrast with regularized stochastic policies.


We will follow the reviewers' suggestion of adding a conclusion, and will discuss the limitations of this work as done above.

[1] Robust reinforcement learning via adversarial training with langevin dynamics, Kamalaruban et al. 2020

[2] Robust Adversarial Reinforcement Learning, Pinto et al. 2017

[3] Action Robust Reinforcement Learning and Applications in Continuous Control, Tessler et al. 2019

[4] Identifiability in inverse reinforcement learning, Cao et al. 2021

[5] A theory of regularized Markov Decision Process Geist et al. 2018

---

### Author Response · Authors · 2022-08-07
**Proposed Conclusion**

Dear reviewer,

Thank you for your constructive feedback. In the final version, we would use the additional page to add the conclusion we propose below. The goal is to summarize the contribution and state the limitations of the setting mainly, the entropy regularized expert. We welcome any suggestion from your side about the following conclusion draft.

Best, Authors

**Proposed Conclusion**

In this paper, we analyze conditions that guarantee identifiability of the reward function (up to an additive constant) from multiple observed experts maximizing the same reward and facing different transition dynamics. This hence allows us to train optimal policies in any other environment sharing the same reward with the environments of the observed experts. On the other hand, in order to only generalize to a given environment, such strong reward identification is not required, and we provide a milder necessary and sufficient condition for it. We also provide identifiability results in a variety of settings, i.e., linearly parameterized reward, approximated transition matrices, observation of any number of experts, as well as a non-identifiability result in the presence of exogeneous variables.

The main limitations of this work are the following:

- **Observing experts in different environments:** We saw that observing a single expert in one environment cannot lead to reward identification in our setting. We hence need to observe at least two experts acting in different enough environments. To motivate this assumption, note that varying environments are ubiquitous in RL, in particular in Robust RL which deals with the training of experts that perform well in different environments, where the transition dynamics can vary to some extent. It is hence rather common to consider that the transition dynamics of a given environment can change. This was studied, e.g., in [1,2,3], where the authors considered different Mujoco environments with varying friction coefficients, or object masses, which influence the dynamics. Also, instead of observing different experts in different environments, we could imagine that we observe a single expert in a single environment that varies over time (but with fixed reward), and that the expert adapts to these changes. Such observations would provide us optimal actions in environments with different transition dynamics, and thus our results would apply. This is of particular interest in economics/finance where the environment is in constant evolution.

- **Assuming entropy regularized experts:** When observing real world data, we have to face the fact that humans do not follow this idealized mathematical model. However, it turns out that our results still hold for the more general class of regularized MDPs [4] where we replace the entropy with any strongly convex function. Whether the flexibility in the choice of the regularizer allows to better capture real-world behaviors is an open question.


[1] Robust reinforcement learning via adversarial training with langevin dynamics, Kamalaruban et al. 2020

[2] Robust Adversarial Reinforcement Learning, Pinto et al. 2017

[3] Action Robust Reinforcement Learning and Applications in Continuous Control, Tessler et al. 2019

[4] A theory of regularized Markov Decision Process Geist et al. 2018

---

> ### Comment · Reviewer_Pbuu · 2022-08-08
> **Response to proposed conclusion**
>
> Thank you for this draft of the proposed conclusion, I think it greatly helps in properly positioning this paper's contributions, as well as providing readers with a clearer takeaway.

---

> > ### Author Response · Authors · 2022-08-09
> > **Thanks**
> >
> > Thank you for the positive feedback. We will add this conclusion in the final version.
> >
> > Best,
> > Authors

---

### Meta-Review · Area_Chair_wVs7 · 2022-08-27

**Recommendation:** Accept
**Confidence:** Less certain

**Metareview:**

The paper provides an investigation of conditions for recovering the reward function up to a constant from multiple experts. While the assumption that observations from multiple (entropy regularized experts) acting in different environments is quite strong, the authors did a good job in justifying and further explaining the setting in the rebuttal. While the paper is incremental, I agree with the reviewers that the paper is solid and interesting.

**Award:**

No

---

### Decision · Program_Chairs · 2022-09-14

Accept